# Different Exercise Types Produce the Same Acute Inhibitory Control Improvements When the Subjective Intensity Is Equal

**DOI:** 10.3390/ijerph19159748

**Published:** 2022-08-08

**Authors:** Laura Carbonell-Hernandez, Juan Arturo Ballester-Ferrer, Esther Sitges-Macia, Beatriz Bonete-Lopez, Alba Roldan, Eduardo Cervello, Diego Pastor

**Affiliations:** 1Sports Research Center, Department of Sport Sciences, Miguel Hernández University of Elche, 03202 Elche, Spain; 2Department of Psychology, Miguel Hernández University of Elche, 03202 Elche, Spain

**Keywords:** aging, cognitive function, Stroop, physical exercise, exercise type

## Abstract

Twenty-eight active older people (67.19 ± 4.91 years) who engaged in physical exercise activity twice a week were recruited to participate in a counterbalanced experimental protocol. The participants performed three different exercise sessions on three different days, one based on aerobic activities, one based on strength exercises with elastic bands, and one based on stationary balance games. During all three sessions, they were encouraged to maintain a moderate subjective intensity (5–6 on the RPE10 scale), and their heart rate was recorded. In addition, all of the participants took a digital version of the Stroop test before and after each session. The study aimed to compare the acute cognitive impacts of different types of exercise sessions in older adults. The participants’ heart rate differed between the exercise sessions, but they maintained the RPE intensity. There was a significant improvement in inhibitory control (Stroop test) after all sessions, with no differences between exercise sessions. Moreover, some participants agreed to be genotyped to record the single nucleotide polymorphism of BDNF rs6265. There were no differences between Val/Val and Met carriers at the beginning or end of the exercise sessions. The present study showed similar cognitive improvements with different exercise type sessions when the subjective intensity was maintained.

## 1. Introduction

Aging is a global process that implies the deterioration of different human systems over time [1]. The central nervous system is one of the most affected systems during the aging process [2]. The impact of aging on the brain produces cognitive function impairment [3]. However, the evidence promotes physical exercise to reduce cognitive impairment and even improve cognitive function during aging [4]. For years, scientific research has been studying the frequency, intensity, time, type (FITT) variables that modulate the impact of physical exercise on cognition [5].

Cognitive function is a concept that integrates a large number of cognitive domains. For example, regarding physical exercise, executive functions are a common domain studied in the literature and include variables such as work memory, attention, inhibition, and cognitive flexibility [6]. Moreover, these variables are particularly relevant during learning [7]. In addition, selective attention has been defined as the skill of focusing attention and avoiding distractions, and it has been widely studied due to its importance in the academic environment [8].

The benefits of physical exercise for cognitive function consist of both acute benefits after single sessions [9] and chronic improvements after prolonged exercise programs [5]. Different hypotheses explain acute and chronic cognitive improvement with physical exercise. The increase in brain blood flow, neurotrophic factors such as brain-derived neurotrophic factor (BDNF), the reduction in cortisol, or the control of chronic diseases such as hypertension or diabetes can explain the cognitive improvements [10]. In addition, different studies have investigated the effect of the intensity of exercise on the production of cognitive improvements [11,12,13,14,15,16,17]. Moreover, different studies have evaluated the different types of exercise. These studies have focused on aerobic exercise [18], resistance training [9,11], or different coordinative activities such as Tai Chi [19]. One of the most common forms of exercise training to improve health in older adults is multi-component programs, which include strength, aerobic, and balance exercises, and they have shown benefits even in hospitalized [20] and institutionalized [21] older adults. However, insufficient evidence exists about the impact of different exercise types on cognitive improvements. No comparative studies between different types of exercises have been published.

Moreover, BDNF seems to be a critical factor in cognitive function. In humans, BDNF secretion is regulated by the BDNF gene. During development, BDNF functions are concerned with neuronal growth and the survival and differentiation of neurons [22,23]. These characteristics, along with the well-established role of BDNF in synaptic plasticity, are translated into positive functional changes, making BDNF a key protein implicated in the formation and consolidation of memories [24]. This gene presents a single nucleotide polymorphism (SNP) named rs6262 SNP or BDNF Val66Met gene polymorphism. This SNP substitutes a valine (Val) for methionine (Met) in codon 66. The Met SNP has been found to reduce the BDNF secretion [25]. In addition, in humans, Met carriers have shown reduced effects from exercise benefits on declarative memory [26]. The relevance of controlling the BDNF SNP when studying the impact of exercise on cognitive function seems clear.

Consequently, this study aims to compare the acute cognitive impacts of different types of exercise sessions in older adults. Moreover, the BDNF SNP will be controlled for the study. 

## 2. Materials and Methods

### 2.1. Experimental Design

The participants attended the laboratory four times, once a week for four weeks. In their first session, they were informed about and signed the written consent for the research. In addition, they undertook the training protocol with the Stroop test software. Finally, they performed one of the three different exercise sessions in each of the other three sessions. The cognitive test preceded all of the exercise sessions, and the post-test was carried out 10 min after the end of the sessions.

The exercise sessions were performed at 9:30 in the morning on the same day of the week for all of the participants. In addition, all of the participants were aleatorily counterbalanced to control a possible learning bias.

The study was designed in compliance with the Declaration of Helsinki and approved by the University’s Project Evaluation Committee (evaluation code UMH.CID.DPC.02.17).

### 2.2. Participants

Twenty-eight participants (age 67.19 ± 4.91 years) were recruited for the study. All of them were physically independent, and all of them participated in organized physical exercise activities in their leisure time. Unfortunately, only 14 participants agreed to be genotyped to identify their Val66Met SNP.

### 2.3. Genetic Analysis

Saliva samples were collected with the Orange DNA Kit (OG-500 DNA Genotek Inc., Ottawa, ON, Canada). DNA was extracted following the manufacturer’s protocols. In addition, a quantitative real-time StepOne PCR (Applied Biosystem, Waltham, MA, USA) was used to identify the SNP using a previously described protocol [27]. Participants were described if they carried (MET) or not (VAL) one or two methionine nucleotides. Eight participants were “VAL”, and six were “MET” (five VAL/MET and one MET/MET).

### 2.4. Inhibitory Control Test

Computer software adaptation of the Stroop test was used to measure the inhibitory cognitive capacity [28]. The Stroop test is a standard tool to measure cognitive inhibition [29,30]. The tests were performed on 8-inch tablets (Lenovo TB3-850F, Lenovo Group Limited, Beijing, China).

The test has three different conditions to measure cognitive inhibition. The first condition is congruent, and words (“red, blue, yellow and green”) are presented in black ink (Word Condition). The second condition is neutral, and “XXXX” is presented in different colored inks (Color Condition). The last condition is incongruent, and the words are written with non-coincident inks (Word + Color Condition).

In the three conditions, 50 words were presented to the participants. The failures, successes, and time to complete the 50 words were registered.

### 2.5. Exercise Sessions

Three different exercise sessions were used in the research. Each session was designed with different types of exercises (aerobic exercise, strength exercise with elastic bands, and balance exercises). All of the sessions started with the same 10-min warm-up and finished with the same 10-min cool-down. The central part of each exercise session lasted 30 min.

The warm-up and cool-down presented the same exercises but in reverse order. In the warm-up, the participants undertook 2 min of joint mobility (knees, hips, waist, shoulders, and elbows). Then, they walked for 4 min, increasing their speed every minute until they reached an RPE 5–6 (RPE10 scale). To finish, they conducted three aerobic exercises with a 40 s work and 20 s rest period (exercises can be seen here https://www.youtube.com/watch?v=iRwCUlPobmI&list=PLBUVHpzj7dwOAPml-ZZYXkSUyQeLcwPps, accessed on 16 June 2022).

In the cool-down, the same exercises were conducted in the reverse order, encouraging the participants to reduce RPE to 2–3 at the end of the 4 min walk.

The three sessions were developed on two basketball courts.

The balance session included 21 exercises that were increasingly difficult (exercises can be seen here https://www.youtube.com/watch?v=qBV67H-LJ_g&list=PLBUVHpzj7dwMrgUXWbdhzLQ3ktWRUA5_N, accessed on 16 June 2022). The session included three exercise blocks of 8–9 min of work (one minute each exercise with 40 s work and 20 s rest) with 1 or 2 min at the end to rest and rehydrate (10 min each block). The first block included nine walking exercises with a rope. The second and third block included six reach and pulls exercises with or without a pike, and they performed eight exercises (they repeated some exercises twice) with difficulty feeling secure. The participants did not need to perform all of the exercises, as they had to feel secure, so that they could decide not to increase the difficulty and repeat the previous exercise every moment. They were encouraged to maintain the 5–6 RPE.

The aerobic session included three exercise blocks of eight continuous minutes of work with two minutes of rest to rehydrate. The three blocks consisted of 4 min of walking with 5–6 RPE, and 4 min of walking bouncing a basketball ball; they were allowed to throw the ball to the basket only if they walked all of the basketball courts.

The strength session included three exercise blocks of 8 min of work with 2 min of rest to rehydrate. Eight exercises were used and each block was repeated (the exercises can be seen here https://www.youtube.com/watch?v=GJaMvoZBYh4&list=PLBUVHpzj7dwMrvvlDOSJOWx9OWgVQ6UAG, accessed on 16 June 2022). They worked for 40 s and rested for 20 s with each exercise, maintaining the 5–6 RPE.

In all of the sessions, the participants were encouraged to maintain a moderate intensity (5–6 RPE10 scale). They were continuously reminded of this intensity objective during the sessions. At the end of the sessions, all of the participants declared that they had maintained the objective intensity.

A Polar Team 2 Pro System (Polar Electro Oy, Kempele, Finland) was used to control the physiological intensity of the participants during the sessions. Moreover, the participants’ maximal heart rate (HRmax) was estimated with Tanaka’s formula [31] to control the percentage of HRmax (%HRmax).

### 2.6. Statistical Analysis

A repeated-measure (RM) ANOVA 2 × 3 was used to evaluate the pre-post Stroop results of the three sessions and to analyze the effect of exercise (moment pre–post) vs. the type of exercise (aerobic, strength, and balance). The analysis was repeated for the three conditions of the Stroop test. In addition, the Bonferroni post hoc analysis was undertaken to compare the pairs. 

To analyze the BDNF SNP, an RM ANOVA 2 × 3 × 2 was developed to include the two SNP genotypes. A bilateral signification of *p* < 0.05 was fixed. The Mauchly test was conducted to evaluate the sphericity of the data. The effect size was calculated with square partial eta (η²_p_), grouped into low (≤0.01), medium (≤0.06), and large (≤0.14) effects [32]. The data are presented as the media ± standard deviation. The analysis was performed with JASP 0.16 (Eric-Jan Wagenmakers, Department of the Psychological Methods University of Amsterdam, Nieuwe Achtergracht 129B, Amsterdam, The Netherlands).

## 3. Results

### 3.1. Exercise Session Intensity

Despite the RPE encouragement and RPE10, the %HR differed significantly between the different exercise sessions (F(2, 53) = 5.21, *p* < 0.01). That is a necessary consequence of the different exercise types. The aerobic session presented the highest heart rate intensity (67.69% ± 10.46% HRmax), followed by the strength session (62.4% ± 7.69% HRmax). Finally, the balance session presented the lowest heart rate values (57.8% ± 9.39% HRmax).

### 3.2. Stroop Test: Successes, Failures, and Time

The data showed small differences between the successes and failures in the pre–post analysis. There were no significant differences in a *t*-test. Only time indicated the pre–post differences in all of the conditions (*p* < 0.001). As a consequence, the RM ANOVA was performed with the time data.

### 3.3. Stroop Test: Word Condition

There were significant pre–post differences for the time in the Word Condition for all the sessions (F(2, 58) = 10.25, *p* = 0.002, η²_p_ = 0.150), but there were no differences between the types of exercise in the RM ANOVA 2 × 3 (F(2, 58) = 0.76, *p* = 0.472). The post hoc analysis did not find any significant data for the pre–post sessions (Figure 1).

### 3.4. Stroop Test: Color Condition

There were significant pre–post differences for the time in the Color Condition for all of the sessions (F(2, 58) = 10.25, *p* < 0.001, η²_p_ = 0.326), but there were no differences between the types of exercise in the RM ANOVA 2 × 3 (F(2, 58) = 1.298, *p* = 0.281). The post hoc analysis showed significant differences for the pre–post sessions only in the aerobic (PRE vs. POST: M = 4.57, SE = 1.45, t(19) = 3.153, *p* = 0.038) and strength session (PRE vs. POST: M = 5.73, SE = 1.35, t(22) = 4.244, *p* = 0.001) (Figure 2).

### 3.5. Stroop Test: Word + Color Condition 

There were significant pre–post differences for the time in the Word Condition for all of the sessions (F(2, 58) = 17.48, *p* < 0.001, η²_p_ = 0.232), but there were no differences between the types of exercise in the RM ANOVA 2 × 3 (F(2, 58) = 0.17, *p* = 0.847). The post hoc analysis did not find any significant data for the pre–post sessions (Figure 3).

### 3.6. BDNF SNP Influence

The RM ANOVA 2 × 3 × 2 was carried out to analyze the influence of the BDNF SNP in the different pre–post Stroop conditions. Unfortunately, no significant results were obtained for any of the Stroop conditions when we took care of the SNP.

## 4. Discussion

Many articles have been published regarding aerobic exercise and cognitive function [33]. However, there has been an increase in evidence related to other exercise types to improve cognition in the past years such as resistance training and balance/coordinative exercise (e.g., Tai Chi) [34,35,36].

In our study, all of the exercise types produced the same significant cognitive improvements, with no differences between the exercise types. Only in the Color Condition did there seem to be some differences in the post hoc analysis, but the RM ANOVA results did not support these differences.

These results can be determined by exercise intensity. In our study, the HR response differed between the different exercise types, and it was a foreseeable consequence of the exercise types. According to the American College of Sports Medicine (ACSM) guidelines [37], our strength and balance sessions were light (50–63% HRmax) and the aerobic session was of moderate intensity (64–76% HRmax). Previous studies have found greater acute improvements with higher intensity in aerobic [38] and strength training [11].

However, %HRmax is not a good tool for evaluating the intensity in the strength and balance exercises because the HR response is not related to fatigue in these activities. The intermittent nature of strength training, with short efforts interspersed with recovery periods, means that the heart rate response throughout the exercise sessions does not reflect what occurs at the physiological level [39,40]. While the release of hormonal factors and increased neural drive during efforts leads to an almost immediate increase in heart rate, it takes longer for the cardiovascular system to respond [41]. In our study, to maintain a constant intensity independent of the exercise type, we used the subjective perception with RPE10. As a result, all of the participants maintained a moderate intensity (5–6 RPE10) in all of the exercise sessions independently of the exercise type. The subjective intensity integrates the physiological and psychological signals related to the intensity [42] and may be better related to the cognitive response to acute exercise [33]. Therefore, the subjective intensity maintained could be the reason for the study not finding differences between the three exercise types in our research.

Few articles have analyzed different exercise types. However, in a recent meta-analysis, Gallardo-Gómez et al. [43] analyzed different exercise types concerning cognitive improvement with chronic exercise. In their results, resistance training seemed to be better able to improve cognitive function in older people. Our results cannot be compared with this interesting meta-analysis because we analyzed the acute effect of exercise. Our research did not seem to show differences between the exercise types when the subjective intensity was maintained.

Finally, we failed to corroborate the research hypothesis related to the BDNF SNP. In our hypothesis, the BDNF SNP should modulate the cognitive response after an acute exercise session. However, in agreement with other studies [44], in our research, the BDNF SNP did not modify the cognitive improvements after any different exercise session.

Several mechanisms have been proposed to explain the benefits of physical exercise for cognitive function [45]. One of the plausible underpinnings is enhanced hippocampus neurogenesis [46], which, in turn, could be directly linked to BDNF [45]. As above-mentioned, the homonymous gene regulates BDNF secretion, but the levels released could be downregulated in the presence of the Val66Met polymorphism. However, our study has not found any influence of the said polymorphism on the cognitive response following exercise. For acute exercise, some distinct markers have been explored in their relationship to cognitive response such as catecholamines [47] and lactate [48]. Moreover, aside from genetic factors and biological markers, which were investigated for their role in the exercise–cognition relationship, other theories have attempted to explore the latter relationship through the psychological lens. Along this line of thought, exercise-induced arousal could explain an improvement in the processing speed, assessed through the reaction time tasks. However, it is worth noting that the impact of arousal on the central domains of such tasks remains limited [11]. On the other hand, psychological well-being, which is positively affected by physical exercise [49], has been established as a modulator of the learning process and has even been proposed as a determinant of academic achievement in young students [50,51].

## 5. Conclusions

This article shows a cognitive improvement after exercise but no differences between different exercise types. It seems that aerobic, strength, and balance exercises can produce the same improvements if they are performed with the same subjective intensity, independently of the heart rate response. Moreover, BDNF SNP Val66Met does not appear to modulate this response, and all the participants obtained the same benefits regardless of their genetic background.

## 6. Limitations of the Study

This article presents some limitations that must be improved in future research. First, the number of participants was low, particularly for the BDNF SNP analysis, and the data could be non-significant due to the small amount of data. Moreover, some variables that can modulate the cognitive responses in acute exercises such as mood affects [52] were not measured. 

Finally, this study was designed to analyze the differences between three different exercise types, and there was no control group to ensure that the cognitive differences in the pre–post analysis were due to exercise. However, in agreement with the literature and the presence of improvements in the three sessions, a stable response was shown that was probably due to the exercise session.

## Figures and Tables

**Figure 1 ijerph-19-09748-f001:**
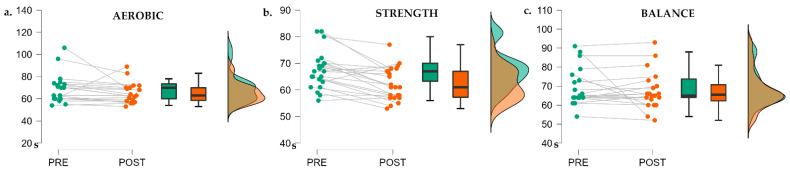
The raincloud plots for the different exercise sessions in the Word Condition. (**a**) Aerobic session. (**b**) Strength session. (**c**) Balance session. Time is measured in seconds (*Y*-axis).

**Figure 2 ijerph-19-09748-f002:**
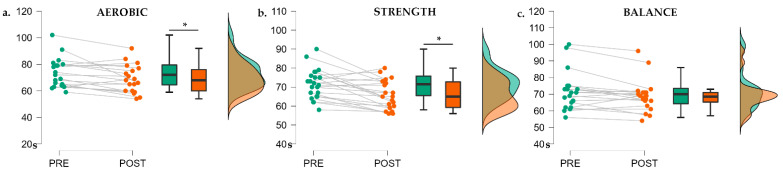
The raincloud plots for the different exercise sessions in the Color Condition. (**a**) Aerobic session. (**b**) Strength session. (**c**) Balance session. Time is measured in seconds (*Y*-axis). * <0.05 in the post hoc analysis.

**Figure 3 ijerph-19-09748-f003:**
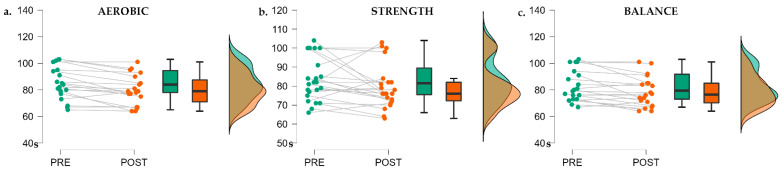
The raincloud plots for the different exercise sessions in the Word Condition. (**a**) Aerobic session. (**b**) Strength session. (**c**). Balance session. Time is measured in seconds (*Y*-axis).

## Data Availability

Not applicable.

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
