# Peer review of "Different Exercise Types Produce the Same Acute Inhibitory Control Improvements When the Subjective Intensity Is Equal"

_ijerph, 2022, doi:10.3390/ijerph19159748_

Round 1
Reviewer 1 Report
Here are my feedbacks.
Abstract
Please provide some statistical values (p value & effect size) of the results.
Introduction
Line 54: Please provide some examples on how BDNF affect cognitive function.
Methods
Line 104: Please provide more details on the warm up and cool down.
Line 107-110: Please provide more details on the exercises. You may consider including a table for this.
Discussion
Line 189-190: Greater improvement in what aspect? And was this after a period of training or acute effect?
Line 192: You mentioned that HR is not related to fatigue in these activities, please provide reference and also define what you mean by fatigue, mental, physical etc.
Author Response
ANSWER TO REVIWER 1
Thanks very much for your review, I hope to resolve all the doubts.
Introduction
Line 54: Please provide some examples on how BDNF affect cognitive function.
Line 56, has been included: “During the development, BDNF functions are concerned with neuronal growth and the survival and differentiation of neurons [22,23]. These characteristics, along with BDNF´s well-established role in synaptic plasticity, are translated into positive functional changes, making BDNF a key protein implicated in the formation and consolidation of memories [24].”
Methods.
Line 104: Please provide more details on the warm up and cool down.
It has been included:
“Warm-up and cool-down present the same exercises but in reverse order. In the warm-up, the participants did 2 minutes of joint mobility (knees, hips, waist, shoulders, and elbows). Then they walk for 4 minutes, increasing speed every minute until they reach an RPE 5-6 (RPE10 scale). To finish, they did three aerobic exercises with 40s work 20s rest period (exercises can be seen here https://www.youtube.com/watch?v=iRwCUlPobmI&list=PLBUVHpzj7dwOAPml-ZZYXkSUyQeLcwPps )
In cool-down, the same exercises were done in the reverese order, encouraging participants to reduce RPE to 2-3 an the end of the 4 minutes walk.”
Line 107-110: Please provide more details on the exercises. You may consider including a table for this.
It has been included:
“The three sessions were developed on two basketball courts.
The balance session included 21 exercises an increasingly difficult (exercises can be seen here https://www.youtube.com/watch?v=qBV67H-LJ_g&list=PLBUVHpzj7dwMrgUXWbdhzLQ3ktWRUA5_N ). The session included three exercise blocks of 8-9 minutes of work (one minute each exercise with 40s work and 20s rest) with 1 or 2 minutes at the end to rest and rehydrate (10 minutes each block). The first block included nine walk exercises with a rope. The second and third block includes six reach and pulls exercises with or without a pike, and they did eight exercises (they repeated twice some exercises) with difficulty feeling secure. The participants do not need to do all the exercises, as they must feel secure, so they can decide not to increase the difficulty and repeat the previous exercise every moment. They were encouraged to maintain the 5-6 RPE
The aerobic session included three exercise blocks of 8 continuous minutes of work with two minutes of rest to rehydrate. The three blocks consist of 4 minutes of walking with 5-6 RPE, and 4 minutes of walking bouncing a basketball ball, they were allowed to throw the ball to the basket only if they walk all of the basketball courts.
The strength session included three exercise blocks of 8 minutes of work with 2 minutes of rest to rehydrate. Eight exercises were used and repeated each block (exercises can be seen here https://www.youtube.com/watch?v=GJaMvoZBYh4&list=PLBUVHpzj7dwMrvvlDOSJOWx9OWgVQ6UAG ). They work 40 s and rest 20 s with each exercise, maintaining the 5-6 RPE.”
Discussion
Line 189-190: Greater improvement in what aspect? And was this after a period of training or acute effect?
Line 190: it has been included “acute” in the sentence. Now: “Previous studies have found greater acute improvements with higher intensity in aerobic [38] and strength training [11]”.
With respect to acute exercise, current evidence suggests that as long as the rest period is introduced after the exercise bout, intense aerobic exercise should lead to the largest improvements in different cognitive domains (Chang et al., 2012)[38]. Moreover, there is also a correlation between %RM in resistance training and cognitive response (Chang et al. 2009)[11].
Line 192: You mentioned that HR is not related to fatigue in these activities, please provide reference and also define what you mean by fatigue, mental, physical etc.
In line 193 has been included: “The intermittent nature of strength training, with short efforts interspersed with recovery periods, means that heart rate response throughout exercise sessions does not reflect what occurs at the physiological level [39,40]. While the release of hormonal factors and in-creased neural drive during efforts leads to an almost immediate increase in heart rate, it takes longer for the cardiovascular system to respond [41].”
We are speaking about the physical fatigue perceived (RPE).

Reviewer 2 Report
The article is original and innovative (use of the Stroop test) and includes psychological aspects together with physical activities that help detect cognitive differences in the pre- and post-analysis. As the authors say in the limitations, they should continue the study increasing the sample, but the current results are good. I suggest that you include the objective in the initial summary.
Author Response
ANSWER TO REVIWER 2
The article is original and innovative (use of the Stroop test) and includes psychological aspects together with physical activities that help detect cognitive differences in the pre- and post-analysis. As the authors say in the limitations, they should continue the study increasing the sample, but the current results are good. I suggest that you include the objective in the initial summary.
Thanks very much for your review, I hope to resolve all the doubts.
It has been included in the summary: “The study aimed to compare the acute cognitive impacts of different types of exercise sessions in older adults.”

Reviewer 3 Report
In this paper, the authors investigated cognitive improvements during three different exercises when the subjective intensity is equal. The topic seems important and useful, and the manuscript was well written. However, this study needs to be improved as follows.
1. The number of participants was too small to give a statistically significant result. The conclusion could be easily changed when adding more samples.
2. In the manuscript, there are no figures to show the types of the three different exercises. The review thinks that the authors could list the exercise types on a table and show and explain them in pictures.
3. The results were only demonstrated. Please add more discussions according to psychological theory to explain the mechanism behind the result.
Author Response
ANSWER TO REVIWER 3
In this paper, the authors investigated cognitive improvements during three different exercises when the subjective intensity is equal. The topic seems important and useful, and the manuscript was well written. However, this study needs to be improved as follows.
Thanks very much for your review, I hope to resolve all the doubts.
- The number of participants was too small to give a statistically significant result. The conclusion could be easily changed when adding more samples.
Undoubtedly, all the studies can be more robust with more participants. We calculated with G-Power (version 3.1.9.6) that we need 25 subjects with an η²p = 0.1 to reach a 0.8 power. In our pot-hoc analysis, we reach a Power of 0.98 (Word Condition) and 0.99 (Color and Word+Color Conditions) in the significant results.
- In the manuscript, there are no figures to show the types of the three different exercises. The review thinks that the authors could list the exercise types on a table and show and explain them in pictures.
The exercise sessions are now described, and there are links to the specific exercises.
- The results were only demonstrated. Please add more discussions according to psychological theory to explain the mechanism behind the result.
At the end of the discussion it has been included:
“Several mechanisms have been proposed to explain the benefits of physical exercise for cognitive function (Norman et al., 2018). One of the plausible underpinnings is enhanced hippocampus neurogenesis (van Praag, 2008), which, in turn, could be directly linked to BDNF (Norman et al., 2018). As mentioned above, the homonymous gene regulates BDNF secretion, but the levels released could be downregulated in the presence of Val66Met polymorphism. However, our study has not found any influence of the said polymorphism on the cognitive response following exercise. For acute exercise, some distinct markers have been explored in their relationship to cognitive response, such as catecholamines (McMorris, 2016) and lactate (Hashimoto et al., 2021). Moreover, aside from genetic factors and biological markers, which were investigated for their role in the exercise-cognition relationship, other theories attempted to explore the latter relationship through the psychological lens. Along this line of thought, exercise-induced arousal could explain an improvement in processing speed, assessed through reaction time tasks. However, it is worth noting that the impact of arousal on the central domains of such tasks remains limited (Chang et al., 2009). On the other hand, psychological well-being, which is positively affected by physical exercise (Cervelló et al., 2014), has been established as a modulator of the learning process and has even been proposed as a determinant of academic achievement in young students (Garcia et al., 2015; Yu et al., 2017).”

Round 2
Reviewer 1 Report
I have no further comments.
Reviewer 3 Report
The authors have properly responded to my comments. No further comments.